# Automated Static Magnetic Cleanliness Screening for the TRACERS Small-Satellite Mission

Cole J Dorman[1,2], Chris Piker[1], David M Miles[1]

[1]Department Physics and Astronomy, University of Iowa, Iowa City, 52242, USA
[2]Department of Climate and Space Sciences and Engineering, University of Michigan, Ann Arbor, 48109, USA

*Correspondence to*: David M Miles (david-miles@uiowa.edu)

**Abstract.** The Tandem Reconnection and Cusp Electrodynamics Reconnaissance Satellites (TRACERS) Small Explorers mission requires high-fidelity magnetic field measurements for its magnetic reconnection science objectives and for its technology
demonstration payload MAGnetometers for Innovation and Capability (MAGIC). TRACERS needs to minimize the local magnetic noise through a magnetic cleanliness program such that the stray fields from the spacecraft and its instruments do not distort the local geophysical magnetic field of interest. Here we present an automated magnetic screening apparatus and procedure to enable technicians to routinely and efficiently measure the magnetic dipole moments of potential flight parts to determine whether they are suitable for spaceflight. This procedure is simple, replicable, and accurate down to a dipole moment of $1.59 \times 10^{-3}$ N m T-1. It
will be used to screen parts for the MAGIC instrument and other subsystems of the TRACERS satellite mission to help ensure magnetically clean measurements on-orbit.

## 1    Introduction

The Tandem Reconnection and Cusp Electrodynamics Reconnaissance Satellites (TRACERS) are twin spacecraft that will be launched into a polar Earth orbit and transit the geomagnetic cusps to study magnetic reconnection (Kletzing, 2019). TRACERS
will make high-fidelity measurements of the local magnetic field using a scientific fluxgate magnetometer as part of the multi-instrument science package and a hosted do-no-harm technical demonstration instrument (Miles et al., 2021) called MAGnetometers for Innovation and Capability (MAGIC) designed to make magnetic measurements without relying on the legacy fluxgate ring-cores (Greene et al., 2022; Miles et al., 2019, 2022). The local in-situ geophysical magnetic field will be contaminated by stray magnetic fields created by the spacecraft's subsystems and onboard scientific instruments. Minimizing this magnetic
contamination is therefore critical to making high-fidelity magnetic field measurements.

The TRACERS mission is developing a magnetic cleanliness program similar to that used by comparable previous missions (e.g., Kuhnke et al., 1998; Ludlam et al., 2009; Matsushima et al., 2010; Narvaez, 2004; de Soria-Santacruz et al., 2020). Currently, the mission design places an upper limit on the total stray field, as observed by the science magnetometer mounted at the tip of a deployable boom, so that the stray field will not degrade the magnetic field measurements below the mission requirements of less
than 100 nT total from all sources. As the mission is developed, this total will be allocated out to the various subsystem and monitored to ensure compliance.

The magnetic field that an object generates is dependent on the strength, orientation, and order of the source. A magnetic dipole moment is typically a good approximation for magnetic field generation when measured far from the source. Figure 1 illustrates how an example magnetic contamination source in the body of the spacecraft creates a stray magnetic field, modeled as a simple
dipole, where the intensity and direction measured by the fluxgate sensors vary depending on the relative orientation and distance from the object. Notably, the inboard MAGIC magnetometer sensor and the outboard MAG science payload magnetometer will

experience different amplitudes of this stray field (ΔB) creating opportunities to use signal processing to potentially identify and mitigate the stray field while preserving the target geophysical field (e.g., Finley et al., 2022; Ness et al., 1971; Neubauer, 1975; Sen Gupta and Miles, 2022; Sheinker and Moldwin, 2016). The screening apparatus presented in this manuscript makes use of the same dependence on the distance from the source. We measure the stray field from an object at two or more distances and fit the dependence with distance to estimate the dipole moment.

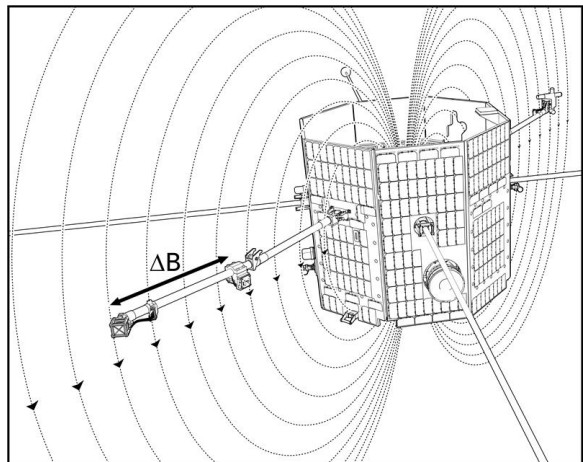

**Figure 1: The two TRACERS magnetometers are deployed on a boom and experience the stray field from the spacecraft at different intensities. We use this same dependence on distance to screen potential components for magnetic cleanliness.**

The two magnetometer sensors on TRACERS are deployed away from the spacecraft on a 1 meter boom. This means that, for example, a dipole moment of 0.05 N m $T^{-1}$ would generate a 10 nT stray field at the outboard magnetometer sensor. This will be used as an example screening standard throughout this manuscript, as the final thresholds are being determined and allocated by the TRACERS magnetics control board. With this magnetic threshold as the standard, all higher order magnetic moments are negligible and relevant calculations are needed only from the dipole moment. We designed, implemented, and tested a magnetic screening process intended to be run on every component of MAGIC and TRACERS using a screening apparatus that is simple, replicable, and reliable. This newly developed magnetic screening apparatus will help ensure sufficiently small stray magnetic noise for the sensitive data collection required by TRACERS and a successful technical demonstration of the MAGIC magnetometer.

## 2    Methodology

As an object with an unknown dipole moment rotates about itself, a magnetometer placed at an arbitrary distance away from the object will measure sinusoidal magnetic field components as the magnetic dipole moment points towards and away from the magnetometer. Regardless of the dipole moment's unknown orientation (excepting the unique case where the dipole is parallel to the axis of rotation which is discussed later), the magnetometer will read maxima and minima magnetic field components and the relative modulation will occur at different strengths in each measured magnetic vector component depending on the orientation of the dipole. Spin-modulating an object's stray magnetic field places it at a specific and constant frequency allowing it to be separated from other local noise sources – many of which occur at or near DC or at variable frequencies. The magnetic field will decay with distance so magnetometers farther from the rotating object will see similar modulation at a reduced amplitude. Figure 2 illustrates the concept of the screening apparatus, based on this effect, where sensors at two different distances see the spin modulated field at different amplitudes depending on the separation distance of the object to be tested and the magnetometer sensor.

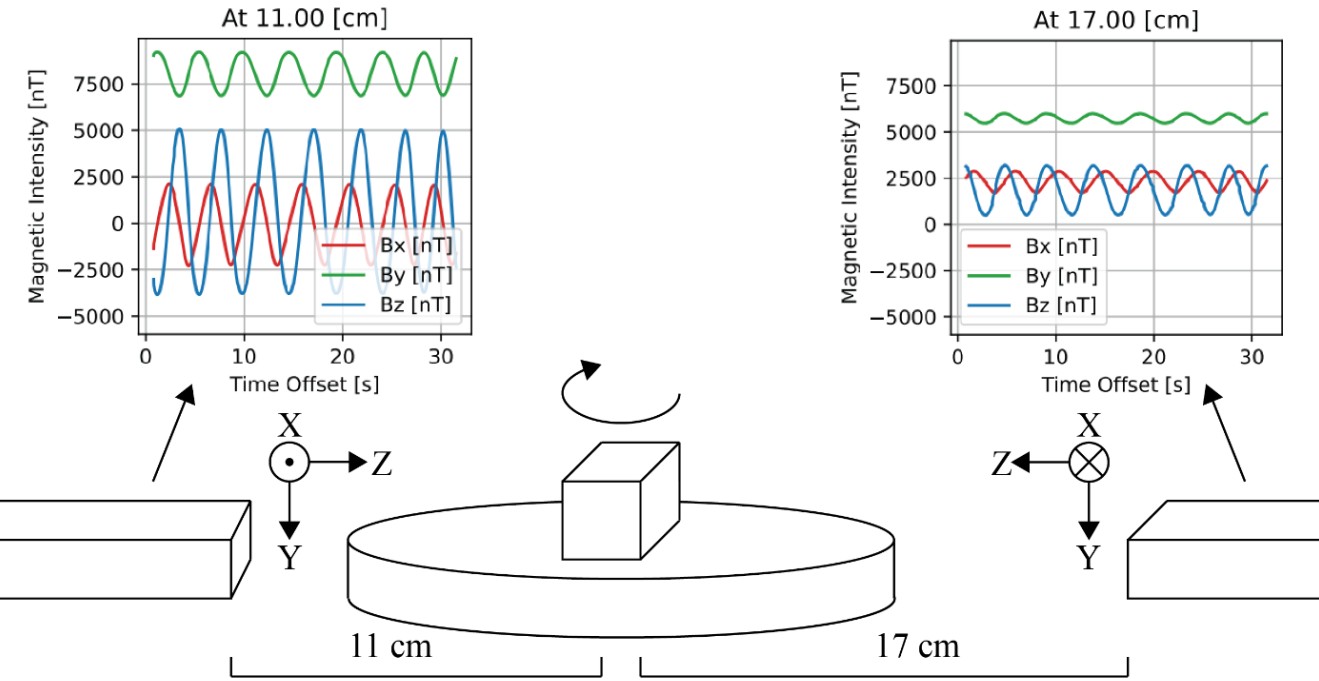

**Figure 2: Schematic of automated magnetic screening procedure and apparatus. The object under test is rotated at a constant velocity and the spin-modulated stray magnetic field is measured at different amplitudes by two or more sensors at different distances from the object.**

## 2.1    Calculations

We use a discrete Fourier transform (DFT) to recover the DC value of the measured stray magnetic field components that have been modulated into an AC sinusoidal magnetic field by the object's rotation (Heinzel et al., 2002). The DFT takes the uniformly spaced time-series samples of magnetic field component readings, then transforms the data to an equally spaced summation in frequency space using Equation 1:

$$y = \sum_{k=0}^{N-1} x_l e^{-2\pi i nl/N} , \qquad n = 0 \ldots N\text{-}1 \tag{1}$$

The modulated field can be described as a sum of sinusoidal basis functions. From this, $x_k$ is our input vector of $N$ uniformly spaced samples. Since the time series from the input signal is always real, the output $y$ is real as well. Increasing test time improves the magnitude determination in the frequency domain due to more cycles sampled, but quickly has diminishing returns as the number of objects to be screened increases. Increasing rotation speed can increase the number of cycles sampled in a set amount of time, but it can lead to lost data as the AC magnetic field observed approaches the sampling magnetometer's Nyquist frequency. Screening times and sampling rates chosen with this in mind are discussed in Section 2.2, Screening Process. Figure 3 shows the time series from the two sensors transformed into frequency space showing a distinct amplitude peak corresponding to rotational frequency of the object under test.

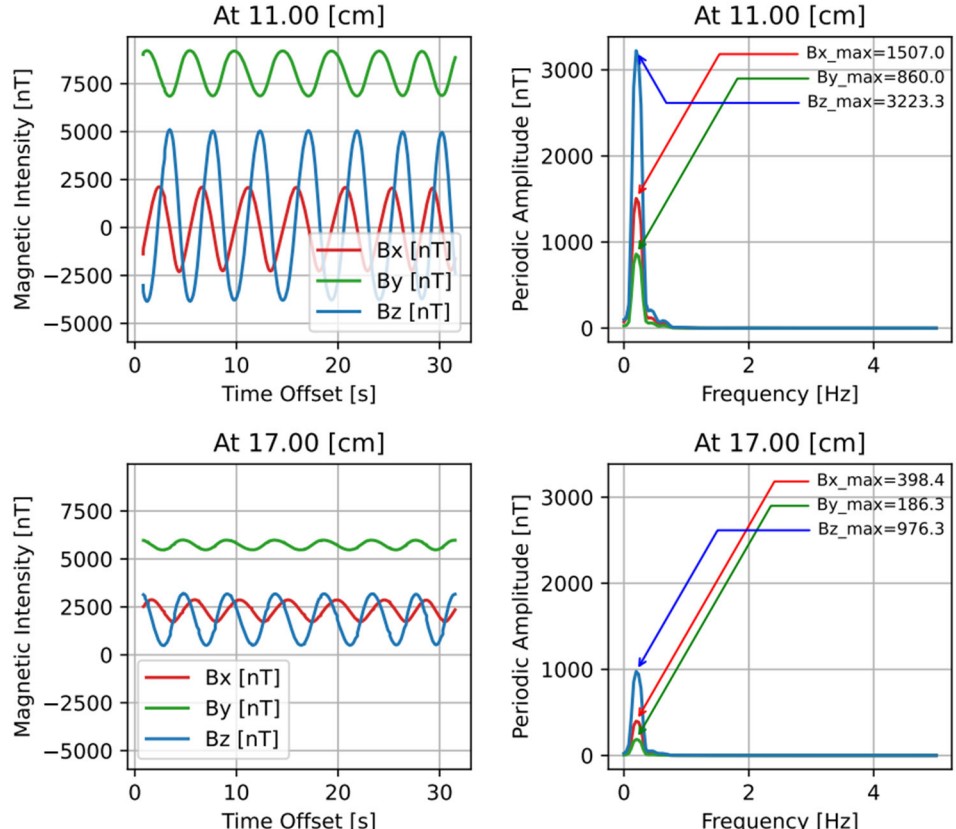

**Figure 3: A discrete Fourier transform is used to convert the modulated magnetic field component readings from Figure 2 into frequency space to isolate and quantitatively measure the spin-modulated field.**

Since the DFT divides frequency space exactly and real-world rotations are not spectrally ideal, a flattop window is used to modify the time-series input signal $x_k$. Without a flattop window, slight changes in rotational frequency could disperse our target spin-modulated signal across multiple frequency bins in the DFT and degrade our estimate of the magnetic field component. A flattop window (D'Antona and Ferrero, 2005) is used to improve the accuracy of amplitude measurement at the expense of reduced frequency resolution (which is irrelevant in this application). Due to the flattop window taking averages of magnetic intensity peak-to-peak, it finds a periodic intensity of the sinusoidal peak divided by √2. The flattop modifies the DFT by:

$$x_l' = x_l * \omega \tag{2}$$

$$\omega = a_0 - a_1 \cos\frac{2\pi n}{N} + a_2 \cos\frac{4\pi n}{N} - a_3 \cos\frac{6\pi n}{N} + a_4 \cos\frac{8\pi n}{N} \tag{3}$$

Here, *a* is a series of constants specific to the flattop algorithm. We use SciPy's implementation of Welch's method of overlapping periodograms to produce an averaged power spectrum with reduced amplitude noise, again at the expense of frequency resolution. A linear power spectrum is taken from the DFT frequency space to measure the magnetic field components. A linear power spectrum is used because the objects spin generates a coherent single frequency signal whose amplitude we need to measure whereas a power spectral density transform normalizes across spectral width and is suitable for broadband incoherent power. Since the rotation is at a constant frequency, the amplitude of a linear power spectrum at the rotational frequency gives the amplitude of the spin-modulated magnetic field from the object directly.

Once the magnetic field components are found, the next step is to calculate the magnetic dipole moment. For a pure dipole field, the magnetic vector is provided by Griffiths (2017) as:

$$\mathbf{B} = \frac{\mu_0 m}{4\pi r^3}(2\cos\theta\, \mathbf{r} + \sin\theta\,\boldsymbol{\vartheta}) = \frac{\mu_0 m}{4\pi r^3}(3\cos\theta\sin\theta\cos\varphi\, \boldsymbol{i} + 3\cos\theta\sin\theta\sin\varphi\, \mathbf{j} + (2\cos^2\theta - \sin^2\theta)\mathbf{k}) \tag{4}$$

Where $B$ is the magnetic field, $\mu_0$ is the vacuum permeability constant, $m$ is the dipole moment, $r$ is distance, $\theta$ is the dipole's angle from the $z$ axis, and $\varphi$ is the dipole's angle from the $x$ axis. Vectors $\mathbf{r}$ and $\boldsymbol{\vartheta}$ denoted radial and azimuthal units in spherical coordinates, and vectors $\mathbf{i}$, $\mathbf{j}$, and $\mathbf{k}$ denote longitudinal, lateral and normal units in cartesian coordinates. Since we are assuming B is a far-field measurement, eq. 4 is sufficient to determine the dipole moment. Also, since the magnetic field vector aligns with the dipole moment vector for large distances, we can use the angle of the magnetic field from the z and x axis to find the dipole moments angle from the $z$ and $x$ axis.

$$\theta = \frac{(B_i,\ B_j,\ B_k) \cdot (\mathbf{k})}{\left|B_i,\ B_j,\ B_k\right| \cdot |\mathbf{k}|}, \qquad \varphi = \frac{(B_i,\ B_j,\ B_k) \cdot (\mathbf{i})}{\left|B_i,\ B_j,\ B_k\right| \cdot |\mathbf{i}|} \tag{5}$$

The dipole moment $m$ can be solved for now that the angles $\theta$ and $\varphi$ for equation 4 are known and the distance variable $r$ is the distance of the magnetometer away from the object. If the calculated dipole moment is less than its allocation then the measured object would be considered suitable to go on the spacecraft. The far-field assumption of B relies heavily on the distance of the measuring sensors from the screening object. If the sensor is at least 5 times farther away from the centered screening objects characteristic radius, the far-field assumption holds (Bansal, 1999)

## 2.2 Screening Process

Test objects are rotated by centering them on a magnetically clean Delrin screening plate on ceramic bearings with paddles to catch a flow of dry nitrogen. This allows the object to be rotated at a constant rate without the use of an electric motor that would generate a variable magnetic field that could contaminate the measurement. It is important that the object on the plate is centered. If misaligned, the object will move towards and away from the screening magnetometers potentially artificially increasing the apparent magnitude of the spin-modulated magnetic field. Notably, this would give an artificially large (conservative) estimate of the dipole moment that could potentially be revised down by retesting with a more accurate centering if required. If there is a dipole offset from the geometric center of the object, it will add multi expansion terms to the magnetic scalar potential as elaborated in Appendix A. However, the distances we are measuring the magnetic fields are much larger than that of the length of the dipole and hence multipole terms fall off quickly enough to be negligible for the scope of this project. The plate is centered in a magnetic shield to reduce, but not completely remove, the background magnetic fields from the Earth and other local magnetic noise sources such as elevators, that can act as cofounders. Shielding may reduce induce fields of a test object however overall, greatly improves accuracy by eliminating large noise sources

A technician places an object to be screened in the center of the rotating plate and increases the flow of dry nitrogen until the object is spinning at a constant rate of around 0.1-0.6 Hz. A software program then commands the magnetometers to collect magnetic field component time series data on the object for a set period, typically 30 seconds. The sampling rate must be much larger than the rotation rate. By default, the software samples at 10 Hz, though this can be increased if necessary. The technician then changes the object's orientation 90° and rescreens the object. This is done in the case that the unknown dipole moment is parallel to the axis of rotation. In that case, the time series of magnetic field component data would appear as if the object were unmoving or had no stray field since the dipole is spin axis symmetric. The technician then takes the larger calculated dipole moment value of the two orientations. After the object is done screening, the technician takes the item off the plate.

## 3    Hardware and Software of the Magnetic Screening Apparatus

### 3.1    Hardware

The magnetically clean rotating screening plate was constructed at the University of Iowa. A flow of dry nitrogen drives a Delrin paddle wheel on ceramic bearings providing non-magnetic rotation. It, along with 2+ Twinleaf VMR magnetometers for data collection, is centered in a 40 x 40 x 40 cm cubic mumetal magnetic shield, which reduces the effect of local magnetic interference. Prior to collecting data, the firmware serial number for each VMR is recorded so that distance values can be automatically associated with each sensor.

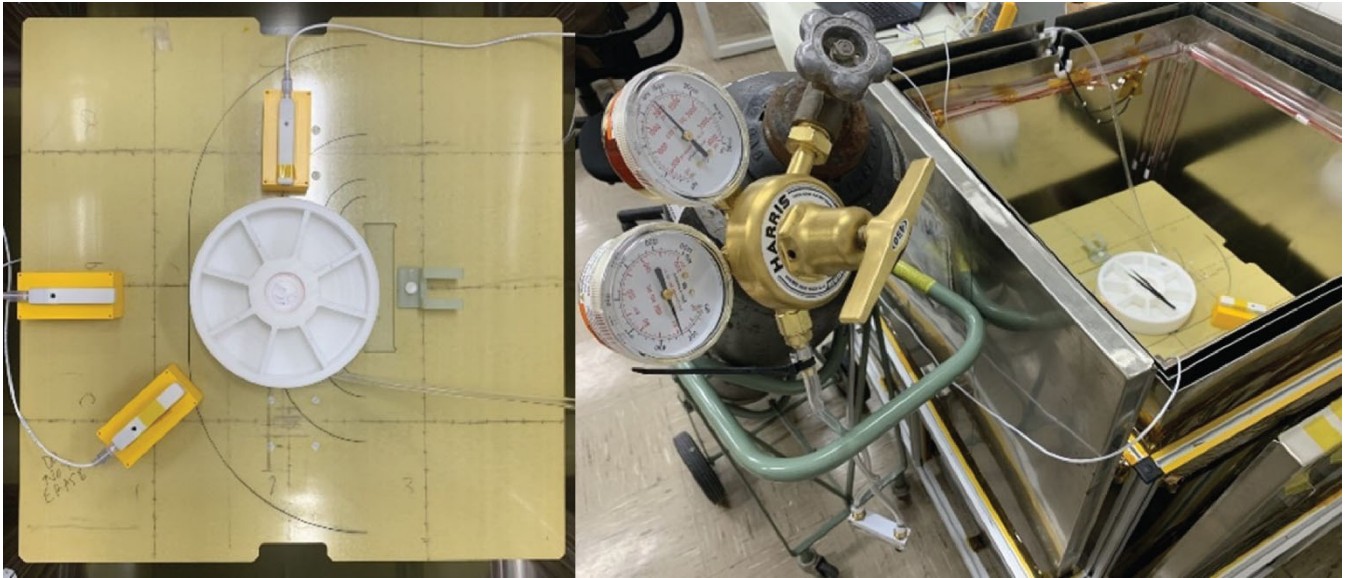

**Figure 4: The magnetic screening apparatus showing (left) a top-down view of the screening plate and Twinleaf VMR magnetometers and (right) a full apparatus view including cylinder of dry nitrogen gas powering object rotation, the ~1x1x1 m magnetic shield, and screening plate.**

### 3.2    Software

The Python new module *magscreen* (https://pypi.org/project/magscreen/) automates magnetic screening by acquiring data from multiple sensors, completing the Fourier analysis, fitting the dipole, and preparing a .pdf report. At the time of publication *magscreen* requires the use of TwinLeaf VMR sensors but could be adapted for other equipment if desired. It provides top-level entry points to automate the data collection, processing, and reporting tasks for ease of use. From each attached magnetometer it reads a 3-axis vector field versus time and transforms this data into magnetic field component intensity at the screening plate's rotational frequency. The worst-case stray magnetic field of an object to be used in-flight is determined from the newly calculated dipole moment and distance $r$ of the magnetometer from the center of the screening plate and an assumed worst-case angle $\theta$ of $\pi/2$. These worst-case fields are then fit into SciPy's curve fit algorithm where a best fit parameter dipole moment is extracted and compared to TRACERS' magnetic cleanliness screening standards. Objects with too large a dipole moment are flagged for further analysis and potential replacement.

The error for the curve fit is estimated from several sources. The Twinleaf VMR magnetometer has an intrinsic measurement error and there is a horizontal measurement error from centering the object and placing the magnetometer sensors, which both create the estimated combined error. There is a vertical alignment error as some objects have non-trivial height which offsets them from the plane of the magnetometer sensors which is set by a finite set of plastic mounts, which is likely quite small and cannot be quantified

in this screening process. These contribute to the error on each calculation of the worst-case fields. Figure 5 shows a best fit dipole based on measurements taken from three sensors at different distances from an object being screened.

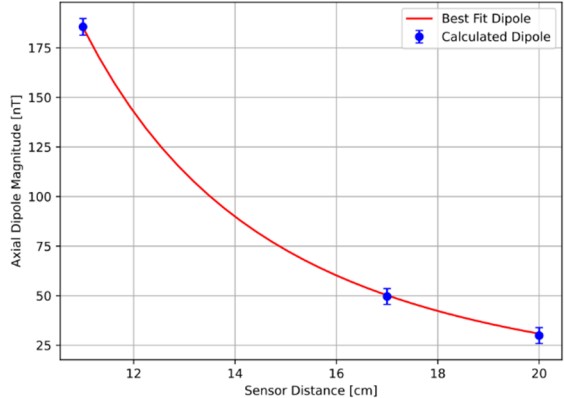

Figure 5: Example magscreen output, a best fit dipole derived from dipoles calculated at each distance.

165 **4    Validations**

The accuracy of the screening process was validated by building and characterizing a reference solenoid shown in Figure 6. The solenoid was driven using a 3 V button cell battery and the magnetic screening procedure yielded a dipole estimate of $3.53 \times 10^{-2} \pm 4.51 \times 10^{-4}$ N m T$^{-1}$ at 0.03 A of current. The magnetic field components were then measured independently at an arbitrary distance along the dipole axis and used to calculate the magnetic dipole following Griffiths (2017):

$$m_k = B_k \frac{2\pi}{\mu_0} R^2 L \left[ \frac{r + L/2}{\sqrt{\left(r + L/2\right)^2 + R^2}} - \frac{r - L/2}{\sqrt{\left(r - L/2\right)^2 + R^2}} \right]^{-1} \tag{6}$$

170    The solenoid had a negligible $B_x$ and $B_y$ component magnetic field and a $B_z$ component of 2073.17 nT at a distance of 15 cm away. The solenoid's radius $R$ was 1.85 cm and had a length $L$ of 1.5 cm. This obtained a dipole moment of $3.56 \times 10^{-2}$ N m T$^{-1}$. The screening procedure and independent calculations values agree within error, demonstrating the robustness of the automated screening apparatus.

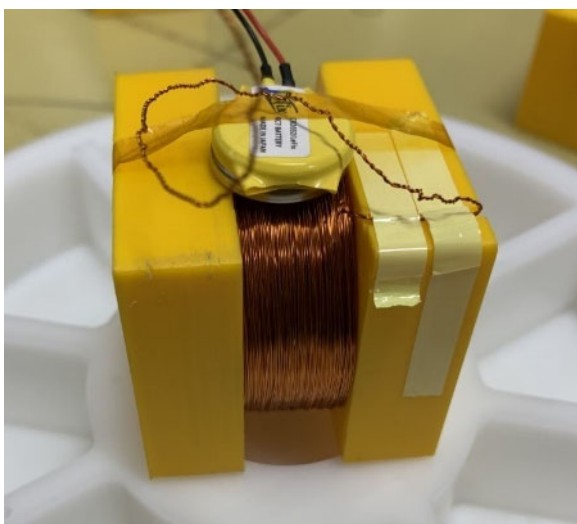

175

**Figure 6: Independently characterized solenoid that was used to validate the automated magnetic screening apparatus and procedure. The wires drawing current from the battery are twisted to minimize additional magnetic fields.**

The dipole moment will vary linearly with current applied. This allows us to validate the sensitivity of the apparatus by iteratively reducing the applied current and creating dipoles ranging from $3.56 \times 10^{-2}$ N m T$^{-1}$ at 0.03 A down to 0 N m T$^{-1}$ at 0 A. The dipole moment has become too small to be resolved by the magnetic screening apparatus when the measured value significantly diverges from field predicted by this current scaling. We therefore take the minimum resolvable dipole to have occurred empirically when the observed moment has an error above 50% compared to the linear fit of dipole as current approaches zero. Figure 7 shows this occurring around $1.59 \times 10^{-3}$ N m T$^{-1}$. Note that, as the stray field of the object get smaller, other noise sources (local and environmental) start to contribute so the dipole tends to be over-estimated when small. Consequently, the output of the screening procedure can be treated as a conservative over-estimate for small stray fields.

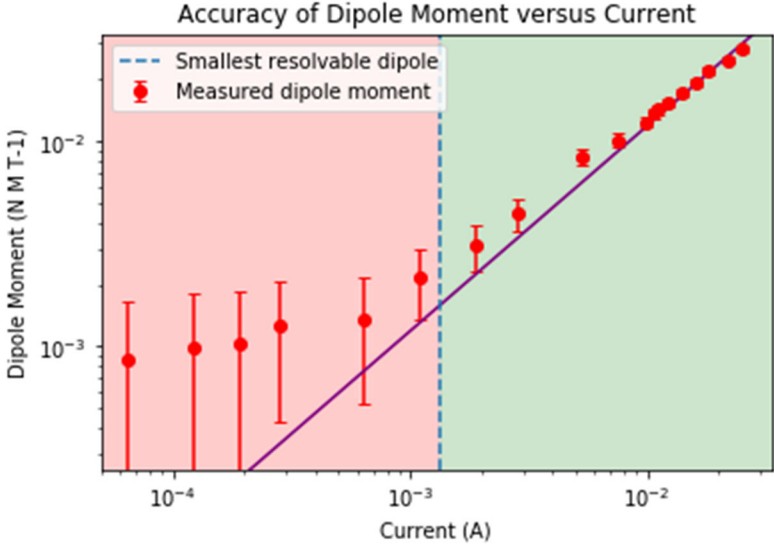

Figure 7: Dipole moment measured by the automated screening apparatus as the current applied to the reference solenoid was reduced to establish the minimum resolvable dipole.

## 5    Conclusion

This automated magnetic screening procedure generates usable results down to dipole moments of $1.59 \times 10^{-3}$ N m T$^{-1}$ using a simple, repeatable process. The procedure involves putting a desired object on the screening plate, rotating it at a constant rate with a flow of dry nitrogen, and running the *magscreen* software which gathers the required magnetic measurements, performs the quantitative spectral analysis, and generates the reports. This apparatus and procedure will help ensure a robust magnetic cleanliness plan for the TRACERS mission and the MAGIC technology demonstration to ensure high-quality, low-noise magnetic field measurements on-orbit.

## 6    Appendix A

If a magnetic dipole moment, M, is present at the center of the screening plate (at the origin), a magnetic potential, $\Psi$, is written as:

$$\Psi(r,\theta,\varphi) = \frac{\mu_0}{4\pi} \frac{M \cdot r}{r^3}$$

Here, r is the position vector and (r, $\theta$, $\varphi$ ) are the spherical coordinates with a magnetic field expressed as:

$$B(r,\theta,\varphi) = -\nabla\Psi(r,\theta,\varphi)$$

When M is present at $r_0$ = ($x_0$, $y_0$, $z_0$), a distance close to the origin, $\Psi$ is expressed as:

$$\Psi(r,\theta,\varphi) = \frac{\mu_0}{4\pi}\frac{M \cdot (r - r_0)}{|r - r_0|^3}$$

$$\Psi(r,\theta,\varphi) = \frac{\mu_0}{4\pi}\frac{M(x - x_0) + M(y - y_0) + M(z - z_0)}{\{(x - x_0)^2 + (y - y_0)^2 + (z - z_0)^2\}^{3/2}}$$

$$\Psi(r,\theta,\varphi) \approx \frac{\mu_0}{4\pi}\left[\begin{array}{l} \dfrac{1}{r^2}\{M_z P_1 + M_x cos\varphi P_1^1 + M_y sin\varphi P_1^1\} \\[2ex] + \dfrac{1}{r^3}\left\{\begin{array}{l} (-M_x x_0 - M_y y_0 + 2M_z z_0)P_2 \\ +\sqrt{3}(M_z x_0 + M_x z_0)cos\varphi P_2^1 \\ +\sqrt{3}(M_z y_0 + M_y z_0)sin\varphi P_2^1 \\ +\sqrt{3}(M_x x_0 - M_y y_0)\cos 2\,\varphi P_2^2 \\ +\sqrt{3}(M_y x_0 - M_x y_0)\sin 2\,\varphi P_2^2 \end{array}\right\} \end{array}\right]$$

$$\Psi(r,\theta,\varphi) = a\sum_{\ell=1}^{2}\sum_{m=0}^{\ell}\left(\frac{a}{r}\right)^{\ell+1}(g_\ell^m \cos m\,\varphi + h_\ell^m \sin m\,\varphi)P_\ell^m(cos\theta)$$

Where P is a Schmidt spherical function of degree l and order m, and a is any unit length. Now using:

$$M_x = \left(\frac{\mu_0}{4\pi}\right)^{-1}a^3 g_1^1 \qquad M_y = \left(\frac{\mu_0}{4\pi}\right)^{-1}a^3 h_1^1 \qquad M_z = \left(\frac{\mu_0}{4\pi}\right)^{-1}a^3 g_1^0$$

We obtain:

$$\begin{bmatrix} -g_1^1 & h_1^1 & 2g_1^0 \\ \sqrt{3}g_1^0 & 0 & \sqrt{3}g_1^1 \\ 0 & \sqrt{3}g_1^0 & \sqrt{3}h_1^1 \\ \sqrt{3}g_1^1 & -\sqrt{3}h_1^1 & 0 \\ \sqrt{3}h_1^1 & \sqrt{3}g_1^1 & 0 \end{bmatrix}\begin{bmatrix} x_0 \\ y_0 \\ z_0 \end{bmatrix} = a\begin{bmatrix} g_2^0 \\ g_2^1 \\ h_2^1 \\ g_2^2 \\ h_2^2 \end{bmatrix}$$

and the position of **M** is given as:

$$x_0 = \frac{a(L_1 - g_1^1 E)}{3H^2} \qquad y_0 = \frac{a(L_2 - h_1^1 E)}{3H^2} \qquad z_0 = \frac{a(L_0 - g_1^0 E)}{3H^2}$$

Where

$$H^2 = (g_1^0)^2 + (g_1^1)^2 + (h_1^1)^2$$
$$L_1 = -g_1^1 g_2^0 + \sqrt{3}(g_1^0 g_2^1 + g_1^1 g_2^2 + h_1^1 h_2^2)$$
$$L_2 = -h_1^1 g_2^0 + \sqrt{3}(g_1^0 h_2^1 - h_1^1 g_2^2 + g_1^1 h_2^2)$$
$$L_0 = 2g_1^0 g_2^0 + \sqrt{3}(g_1^1 g_2^1 + h_1^1 h_2^1)$$
$$E = \frac{L_0 g_1^0 + L_1 g_1^1 + L_2 h_1^1}{4H^2}$$

## 7 Code and Data Availability

The magscreen software and example data used to create the figures in this manuscript are available through the Python Package Index here: https://pypi.org/project/magscreen/

## 8 Author Contributions

C. J. Dorman lead the design, assembly, and test of the screening apparatus and process, analyzed the data, and wrote the manuscript with contributions from all authors. D. M. Miles provided supervision and funding for the project as the lead investigator, guided

the design, and assisted in the interpretation of the results. C. Piker lead the software effort to automate the magscreen data
collection, processing, and report generation of the magscreen software.

## 9 Acknowledgments

This project is based upon work supported by the University of Iowa Physics & Astronomy Charles A. Wert Summer Research Grant, the Iowa Space Grant Consortium under NASA Award No. 80NSSC20M0107, and NASA Contract No. 80GSFC18C0008 administered by Goddard Space Flight Center. The authors thank Drs. Matthew Finley and Sapna Shekhar for their careful
proofreading of this manuscript and their thoughtful comments. The authors thank Christian Hansen and Damion Johnson for designing and fabricating the mechanical components of the screening apparatus.

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
