# Peer review of "Automated Static Magnetic Cleanliness Screening for the TRACERS Small-Satellite Mission"

_EGUsphere, 2022_

## Referee Comment (RC1)

A review of the paper EGUsphere-2022-480

**Automated Static Magnetic Cleanliness Screening for the TRACERS Small-Satellite Mission**

Cole J. Dorman, Chris Piker, and David M. Miles

The magnetic cleanliness program is significant to achieve scientific objectives related to magnetic field measurements. Any magnetic tests are to be carried out for all components and subsystems of engineering and flight models of spacecraft so as to suppress stray magnetic fields. Hence, the authors present an automated magnetic screening apparatus and procedure for the purpose as mentioned above. An object to be tested is put on the center of the magnetically clean plate which can be rotated by a flow of nitrogen. The plate and magnetometers are put in a cube made of mu-metal with very high permeability to magnetically shield them. It is important that not only specialists in magnetic cleanliness but also technicians can carry out such magnetic tests routinely and efficiently.

The subject seems to be appropriate for publication in *Geoscientific Instrumentation, Methods and Data Systems*. Its significance and quality are highly evaluated. However, there are some points to be reconfirmed and clarified. Furthermore, if the software is improved, not only the magnetic dipole moment in an object but also its location can be determined. I do not think that the improvement is very difficult. I therefore require moderate revision. I offer comments below for the authors' consideration of revision.

Line 54, Figure 2, and line 97

It is better to specify the maximum size of an object to be tested. This point is related to an assumed far-field measurement.

Figures 2 and 3

The subplots in Fig. 2 (times series measured by magnetometers at 11 cm and at 17 cm) are identical to the subplots in Fig. 3. This means that the authors can rearrange these figures to one figure.

In the upper-left subplot of Fig. 3, the peak-to-peak amplitude of $B_z$ seems to be about or larger than 4000 nT, but the corresponding periodic amplitude in the upper-right subplot is 3223.3 nT. Is this caused by a flattop window applied to time series? If it is the case, it is better to mention it. By the way, which is better, use of a flattop window or not?

Equation (1)

The subscript $m$ should be specified. Later, $m$ is used as the magnetic dipole moment.

Equations (2) and (3)

The subscript $k$ should be specified.

Equations (4) and (5)

Vectors $\boldsymbol{r}$, $\boldsymbol{\vartheta}$, $\boldsymbol{i}$, $\boldsymbol{j}$, and $\boldsymbol{k}$ should be specified.

" , " between two equations for $\theta = \cdots$ and $\phi = \cdots$ is significant to separate these equations, so that add "comma" in Equation (5).

It is better to use different sign for the inner product of vectors, $\cdot$, and scalar multi-plication.

Lines 106–110

"It is important that the object on the plate is centered." I agree with it. However, a magnetic dipole in the object is not necessarily present at the center of the object. In the same sense, how about the hight of magnetometers against the object? In other words, an offset dipole moment should be taken into account. This suggests that the present method may have any defect. To overcome this point, the authors can determine spherical harmonic coefficients up to degree 2.

As the authors understand, if a magnetic dipole moment, $\boldsymbol{M}$, is present at the center (at the origin), a magnetic potential, $\Psi$, is written as

$$\Psi(r, \theta, \phi) = \frac{\mu_0}{4\pi} \frac{\boldsymbol{M} \cdot \boldsymbol{r}}{r^3},$$

where $\boldsymbol{r}$ is the position vector and $(r, \theta, \phi)$ are the spherical coordinates (it should be noted that definition of $\theta$ and $\phi$ is different from that in the manuscript, in which $\theta$ and $\varphi$ are the dipole's angle from the $z$-axis and that from the $x$-axis, respectively), and the magnetic field is expressed as

$$\boldsymbol{B}(r, \theta, \phi) = -\nabla\Psi(r, \theta, \phi).$$

If $\boldsymbol{M}$ is present at $\boldsymbol{r}_0 = (x_0, y_0, z_0)$ which is not very far from the origin, $\Psi$ can be expressed as

$$
\begin{aligned}
\Psi(r, \theta, \phi) \ &= \ \frac{\mu_0}{4\pi} \frac{\boldsymbol{M} \cdot (\boldsymbol{r} - \boldsymbol{r}_0)}{|\boldsymbol{r} - \boldsymbol{r}_0|^3} \\
&= \frac{\mu_0}{4\pi} \frac{M_x(x - x_0) + M_y(y - y_0) + M_z(z - z_0)}{\{(x - x_0)^2 + (y - y_0)^2 + (z - z_0)^2\}^{3/2}} \\
&\approx \frac{\mu_0}{4\pi} \left[ \frac{1}{r^2}\{M_z P_1 + M_x \cos\phi P_1^1 + M_y \sin\phi P_1^1\} \right. \\
&\qquad + \frac{1}{r^3}\left\{(-M_x x_0 - M_y y_0 + 2M_z z_0)P_2 \right. \\
&\qquad\qquad + \sqrt{3}(M_z x_0 + M_x z_0)\cos\phi P_2^1 \\
&\qquad\qquad + \sqrt{3}(M_z y_0 + M_y z_0)\sin\phi P_2^1 \\
&\qquad\qquad + \sqrt{3}(M_x x_0 - M_y y_0)\cos 2\phi P_2^2 \\
&\qquad\qquad \left.\left. + \sqrt{3}(M_y x_0 - M_x y_0)\sin 2\phi P_2^2\right\}\right],
\end{aligned}
$$

where $P_\ell^m$ is a Schmidt spherical function of degree $\ell$ and order $m$. $\Psi$ can also be written as

$$\Psi(r,\theta,\phi) = a \sum_{\ell=1}^{2} \sum_{m=0}^{\ell} \left(\frac{a}{r}\right)^{\ell+1} (g_\ell^m \cos m\phi + h_\ell^m \sin m\phi)\, P_\ell^m(\cos\theta),$$

where $a$ is any unit length (for the geomagnetic potential, $a$ is the Earth's mean radius), and

$$M_x = \left(\frac{\mu_0}{4\pi}\right)^{-1} a^3 g_1^1, \quad M_y = \left(\frac{\mu_0}{4\pi}\right)^{-1} a^3 h_1^1, \quad M_z = \left(\frac{\mu_0}{4\pi}\right)^{-1} a^3 g_1^0.$$

Then we obtain the following equation,

$$\begin{pmatrix} -g_1^1 & -h_1^1 & 2g_1^0 \\ \sqrt{3}g_1^0 & 0 & \sqrt{3}g_1^1 \\ 0 & \sqrt{3}g_1^0 & \sqrt{3}h_1^1 \\ \sqrt{3}g_1^1 & -\sqrt{3}h_1^1 & 0 \\ \sqrt{3}h_1^1 & \sqrt{3}g_1^1 & 0 \end{pmatrix} \begin{pmatrix} x_0 \\ y_0 \\ z_0 \end{pmatrix} = a \begin{pmatrix} g_2^0 \\ g_2^1 \\ h_2^1 \\ g_2^2 \\ h_2^2 \end{pmatrix}.$$

Hence, the position of $\boldsymbol{M}$ is given as

$$x_0 = \frac{a(L_1 - g_1^1 E)}{3H^2}, \quad y_0 = \frac{a(L_2 - h_1^1 E)}{3H^2}, \quad z_0 = \frac{a(L_0 - g_1^0 E)}{3H^2},$$

where

$$H^2 = (g_1^0)^2 + (g_1^1)^2 + (h_1^1)^2,$$

$$L_1 = -g_1^1 g_2^0 + \sqrt{3}(g_1^0 g_2^1 + g_1^1 g_2^2 + h_1^1 h_2^2),$$

$$L_2 = -h_1^1 g_2^0 + \sqrt{3}(g_1^0 h_2^1 - h_1^1 g_2^2 + g_1^1 h_2^2),$$

$$L_0 = 2g_1^0 g_2^0 + \sqrt{3}(g_1^1 g_2^1 + h_1^1 h_2^1),$$

$$E = \frac{L_0 g_1^0 + L_1 g_1^1 + L_2 h_1^1}{4H^2}.$$

**Lines 125–126**

"It $\cdots$ are centered $\cdots$" would be "It $\cdots$ is centered $\cdots$."

"$\cdots$ a 40 × 40 cm cubic $\cdots$" would be "$\cdots$ a 40 × 40 × 40 cm cubic $\cdots$."

**Lines 144–147**

The authors describe that there is a vertical alignment error as one of errors. This can be reduced if the position of a magnetic dipole moment is simultaneously determined as mentioned above.

**Figure 6**

The red and black cables are likely to be used as power lines. Are they twisted? If it is not the case, such a configuration may cause additional magnetic field, so that they should be twisted. If it is the case, it is better to point out the configuration.

**Equation (6)**

The subscript $k$ should be clearly defined. If $k$ stands for $x$, $y$ or $z$, the left-hand-side of Equation (6) should be $m_k$, where $m = (m_x^2 + m_y^2 + m_z^2)^{1/2}$.

---

## Author Response (AR1)

Referee

Journal of European Geosciences Union

DOI: https://doi.org/10.5194/egusphere-2022-480

Title: Automated Static Magnetic Cleanliness Screening for the TRACERS Small-Satellite Mission

Authors: Cole J. Dorman[1, 2], Chris Piker[1], and David M. Miles[1]

[1]Department Physics and Astronomy, University of Iowa, Iowa City, 52242, USA

[2]Department of Climate and Space Sciences and Engineering, University of Michigan, Ann Arbor, 48109, USA

Dear Referee,

We would like to thank you and for your careful consideration and time in handling our manuscript. We truly believe that the revised manuscript has been significantly improved by your suggestions.

The appendix below details our response to each of the comments. We hope that this revised and resubmitted manuscript addresses these comments appropriately. Please let us know if you have any questions regarding our resubmission. Thank you again for handling our manuscript.

Best Regards,

--

Cole J. Dorman

PhD Pre-Candidate

Department of Climate and Space Sciences and Engineering

University of Michigan, Ann Arbor

Email: cjdorman@umich.edu

Phone: (563) 209-3148

Appendix

Key:

*Italics* - Original Reviewer Comment

**Bold** – Author Response

(Text with Parenthesis) – Changed/Added Text

[[Text with Brackets]] – Large Changes to Manuscript, Not Detailed Here

No modifications – Original Text, Included for Context

*Line 54, Figure 2, and line 97*

*It is better to specify the maximum size of an object to be tested. This point is related to an assumed far-field measurement.*

**RE: Thank you for the comment. We added further clarification in lines 103-105 on the maximum sized objects to be tested:**

"If the calculated dipole moment is less than its allocation then the measured object would be considered suitable to go on the spacecraft. (The far-field assumption of B relies heavily on the distance of the measuring sensors from the screening object. If the sensor is at least 5 times farther away from the centered screening objects characteristic radius, the far-field assumption holds (Bansal, 1999))"

*Figures 2 and 3*

*The subplots in Fig. 2 (times series measured by magnetometers at 11 cm and at 17 cm) are identical to the subplots in Fig. 3. This means that the authors can rearrange these figures to one figure.*

**RE: Thank you for your comment. While the subplots in Fig. 2 do reappear in Fig. 3, it is simply for continuity reasons for the reader. We believe Fig. 2's schematic of collecting sinusoidal magnetic data using a spin modulated tray and Fig. 3's demonstration of using discrete Fourier transform is distinct enough in meaning to be stay separated.**

*In the upper-left subplot of Fig. 3, the peak-to-peak amplitude of Bz seems to be about or larger than 4000 nT, but the corresponding periodic amplitude in the upper right subplot is*

*3223.3 nT. Is this caused by a flattop window applied to time series? If it is the case, it is better to mention it. By the way, which is better, use of a flattop window or not?*

**RE: Thank you for the comment. An appeared change in periodic amplitude would be caused by the flattop window. We added clarification:**

"(Due to the flattop window taking averages of magnetic intensity peak-to-peak, it finds a periodic intensity of the sinusoidal peak divided by $\sqrt{2}$.)"

**The flattop window lowers the frequency resolution of the Fourier transform in order to reduce amplitude noise. We believe we sufficiently explained the strength of using a flattop window over no window at all in lines 81-85:**

"Without a flattop window, slight changes in rotational frequency could disperse our target spin-modulated signal across multiple frequency bins in the DFT and degrade our estimate of the magnetic field component. A flattop window (D'Antona and Ferrero, 2005) is used to improve the accuracy of amplitude measurement at the expense of reduced frequency resolution (which is irrelevant in this application)."

*Equation (1)*

*The subscript m should be specified. Later, m is used as the magnetic dipole moment.*

**RE: Thank you for your comment. We decided to just remove the m subscript. It was not necessary and saves any confusion.**

*Equations (2) and (3)*

*The subscript k should be specified.*

**RE: Thank you for your comment. We changed this subscript to be now a subscript "l", as a k subscript appears unrelated later in the manuscript. The subscript l denotes the summation of variable x from l=0 to infinity and is common enough of mathematical notation that the reader will not need explanation. We also chose to remove the k subscript from ω.**

*Equations (4) and (5)*

*Vectors r, ϑ, i, j, and k should be specified.*

**RE: We added additional information:**

"Where B is the magnetic field, $\mu 0$ is the vacuum permeability constant, m is the dipole moment, r is distance, θ is the dipole's angle from the z axis, and $\varphi$ is the dipole's angle from the x axis. (Vectors r and ϑ denoted radial and azimuthal units in spherical coordinates, and vectors i, j, and k denote longitudinal, lateral and normal units in cartesian coordinates.)"

*" , "between two equations for θ = ⋯ and φ = ⋯ is significant to separate these equations, so that add "comma" in Equation (5).*

**RE: We added a comma in between the two equations in Equation (5)**

*It is better to use different sign for the inner product of vectors, ·, and scalar multiplication.*

**RE: Thank you for your comment. We changed Equation (2) to use a "*" sign for scalar multiplication. Now every time a "·" sign is used, it is only for an inner product.**

*Lines 106–110*

*"It is important that the object on the plate is centered." I agree with it. However, a magnetic dipole in the object is not necessarily present at the center of the object. In the same sense, how about the height of magnetometers against the object? In other words, an offset dipole moment should be taken into account. This suggests that the present method may have any defect. To overcome this point, the authors can determine spherical harmonic coefficients up to degree 2…*

**RE: Thank you for the insightful comment and derivation. We have addressed concerns of a multipole expansion arising from a dipole moment not being geometrically centered below:**

[[ We added an Appendix A after the conclusion to derive the multi expansion terms added to the magnetic scalar potential if the dipole moment is not geometrically centered in the object. The appendix serves to demonstrate if we express the magnetic field as a sum of multipole expansion terms and fit it to data, we can determine the offset and the dipole moment.

The length scale of the dipole, however, is much smaller than the measuring distance, so any higher order multipole terms fall off quickly enough to the point where they're negligible. The negligibility of higher order terms preserves the integrity of the current screening method and is validated in Section 4, Validations. ]]

**We now address a dipole offset in lines 113-116:**

"(If there is a dipole offset from the geometric center of the object, it will add multi expansion terms to the magnetic scalar potential as elaborated in Appendix A. However, the distances we are measuring the magnetic fields are much larger than that of the length of the dipole and hence multipole terms fall off quickly enough to be negligible for the scope of this project.)"

*Lines 125–126*

*"It ⋯ are centered ⋯" would be "It ⋯ is centered⋯."*

*"⋯ a 40×40 cm cubic ⋯" would be "⋯ a 40×40×40 cm cubic⋯."*

**RE: Thank you for the comment. We made the appropriate changes in the manuscript.**

"It, along with 2+ Twinleaf VMR magnetometers for data collection, (is) centered in a 40 x 40 (x 40) cm cubic mumetal magnetic shield"

*Lines 144–147*

*The authors describe that there is a vertical alignment error as one of errors. This can be reduced if the position of a magnetic dipole moment is simultaneously determined as mentioned above.*

**RE: Thank you for the comment. We believe your response to comments on Lines 106-110 is sufficient enough where this critique is no longer an issue.**

*Figure 6*

*The red and black cables are likely to be used as power lines. Are they twisted? If it is not the case, such a configuration may cause additional magnetic field, so that they should be twisted. If it is the case, it is better to point out the configuration.*

**RE: The wires drawing power are twisted and we added an explanation in the Figure 6 caption to better reflect that:**

"Figure 6: Independently characterized solenoid that was used to validate the automated magnetic screening apparatus and procedure. (The wires drawing current from the battery are twisted to minimize additional magnetic fields.)"

*Equation(6)*

*The subscript k should be clearly defined. If k stands for x, y or z, the left-hand-side of Equation (6) should be mk, where $m = (m^2_x + m^2_y + m^2_z)^{1/2}$.*

**RE: Thank you for the insight. The dipole, m, effectively only has a z-component. Since Line 97 now denotes** "(vectors i, j, and k denote longitudinal, lateral and normal units in cartesian coordinates)" **and we removed the k subscript from Equations (2) and (3), the reader is confidently informed the k subscript defines the z or normal coordinate unit. We changed Equation (6) to include a k subscript in the left-hand side of the equation, to address your comment and better reflect the system being measured.**

Referee

Journal of European Geosciences Union

DOI: https://doi.org/10.5194/egusphere-2022-480

Title: Automated Static Magnetic Cleanliness Screening for the TRACERS Small-Satellite Mission

Authors: Cole J. Dorman[1,2], Chris Piker[1], and David M. Miles[1]

[1]Department Physics and Astronomy, University of Iowa, Iowa City, 52242, USA

[2]Department of Climate and Space Sciences and Engineering, University of Michigan, Ann Arbor, 48109, USA

Dear Referee,

We would like to thank you and for your careful consideration and time in handling our manuscript. We truly believe that the revised manuscript has been significantly improved by your suggestions.

The appendix below details our response to each of the comments. We hope that this revised and resubmitted manuscript addresses these comments appropriately. Please let us know if you have any questions regarding our resubmission. Thank you again for handling our manuscript.

Best Regards,

--

Cole J. Dorman

PhD Pre-Candidate

Department of Climate and Space Sciences and Engineering

University of Michigan, Ann Arbor

Email: cjdorman@umich.edu

Phone: (563) 209-3148

Appendix

Key:

*Italics* - Original Reviewer Comment

**Bold** – Author Response

(Text with Parenthesis) – Changed/Added Text

[[Text with Brackets]] – Large Changes to Manuscript, Not Detailed Here

No modifications – Original Text, Included for Context

*Achieving a 100 nT cleanliness at 1 m would be achievable using simpler screening equipment. A magnetometer in a magnetic screen would be sufficient. An even easier way is to use an astatic magnetometer that measures the gradient of the dipole moment directly.*

*The described method can however be used for more strict requirements and resembles the Multi Dipole Model method used eg. in MFSA of IABG, Germany to comply with ECSS-e-hb-20-07a*

**RE: Thank you for the insightful comment. While we certainly recognize there are numerous ways to magnetically screen objects, each with their own merit, this method is novel while being simple, replicable, and reliable. Demonstrated in Section 4, Validations, this method is accurate to have virtue in magnetically screening objects for the MAGIC magnetometer demonstration for NASA's TRACERS.**

*Figure 3 On the data processing*

*The peak-to-peak amplitude of the blue AC trace seems to be 2500 nT while the magnitude in the frequency domain is calculated to 976.3 nT. It seems that the flattop window reduces the peak determination by some 20%. This should be discussed.*

**RE: Thank you for the comment. An appeared change in periodic amplitude would be caused by the flattop window dividing the peak-to-peak by $\sqrt{2}$. This is so we can find the average magnetic field, which is the RMS of the peak-to-peak. We added clarification:**

"(Due to the flattop window taking averages of magnetic intensity peak-to-peak, it finds a periodic intensity of the sinusoidal peak divided by $\sqrt{2}$.)"

*Almost 7 full periods are used in the DFT. Using a few periods may significantly influence the magnitude determination in the frequency domain. Please discuss the trade-off between test time, rotation speed sample rate and signal processing (averaging spectrum).*

**We added a brief discussion on trade-offs:**

"(Increasing test time improves the magnitude determination in the frequency domain due to more cycles sampled, but quickly has diminishing returns as the number of objects to be screened increases. Increasing rotation speed can increase the number of cycles sampled in a set amount of time, but it can lead to lost data as the AC magnetic field observed approaches the sampling magnetometer's Nyquist frequency. Screening times and sampling rates chosen with this in mind are discussed in Section 2.2, Screening Process.)"

*Line 88 "to produce an averaged spectrum":*

*This is not described in the procedure section below. How many spectrums are averaged? What noise reduction is achieved by this?*

**RE: Thank you for the comment. The language of 'average spectrum' was meant to convey averaged periodograms to find linear power spectrum. Welch's method of power spectrum estimation reduces amplitude noise in exchange for decreased frequency resolution, which is only beneficial as the exact rotating frequency is irrelevant and the magnetic periodogram amplitude is desired for calculations. The argument for linear power spectrum is discussed later in the paragraph. To clear up confusion, we added:**

"We use SciPy's implementation of Welch's method of overlapping periodograms to produce an averaged (power) spectrum with reduced amplitude noise, again at the expense of frequency resolution."

*Equation 4*

*A figure describing the reference frame the dipole moment and the dipole field would help.*

**RE: Thank you for the comment. We believe we sufficiently described the relationship between reference frame and dipole moment in Lines 86-87:**

"Where B is the magnetic field, $\mu 0$ is the vacuum permeability constant, m is the dipole moment, r is distance, θ is the dipole's angle from the z axis, and $\varphi$ is the dipole's angle from the x axis".

**We address your concerns with relationship between dipole moment and dipole field in the next comment.**

*Line 98 "the magnetic field vector completely aligns with the dipole moment vector"*

*Is that true? I guess the direction of the dipole moment can be determined by the measurements performed but the field vector is not always aligned with the dipole moment vector.*

**RE: Thank you for the comment. At the boom distance of 1 meter, higher order magnetic moments will be far smaller in magnitude than the dipole moment. Due to this fact, we assume higher order moments are negligible and the magnetic field generated is only from the dipole moment. Assuming this then the magnetic field would align with the dipole moment. The negligibility of higher order terms preserves the integrity of the current screening method and is validated in Section 4, Validations. We added clarification in the Introduction:**

"0.05 N m T-1 … will be used as an example screening standard throughout this manuscript, as the final thresholds are being determined and allocated by the TRACERS magnetics control board. (With this magnetic threshold as the standard, all higher order magnetic moments are negligible and relevant calculations are needed only from the dipole moment.)"

*Line 110 "The plate is centered in a magnetic shield to reduce, but not completely remove, the background magnetic fields"*

*The shielding also reduces induced fields from the test object.*

**RE: Thank you for the comment. To address this concern, we have added:**

"The plate is centered in a magnetic shield to reduce, but not completely remove, the background magnetic fields from the Earth and other local magnetic noise sources such as elevators, that can act as cofounders. (Shielding may reduce induce fields of a test object however overall, greatly improves accuracy by eliminating large noise sources.)"

*Line 106 "It is important that the object on the plate is centered"*

*It is important that the apparent magnetic moment of the test object is centered. This is together with the apparent height of the dipole an error source and should be evaluated.*

**RE: Thank you for the insightful comment. We have addressed concerns of a multipole expansion arising from a dipole moment not being geometrically centered in a new Appendix below:**

[[ We added an Appendix A after the conclusion to derive the multi expansion terms added to the magnetic scalar potential if the dipole moment is not geometrically centered in the

object. The appendix serves to demonstrate if we express the magnetic field as a sum of multipole expansion terms and fit it to data, we can determine the offset and the dipole moment.

The length scale of the dipole, however, is much smaller than the measuring distance, so any higher order multipole terms fall off quickly enough to the point where they're negligible. The negligibility of higher order terms preserves the integrity of the current screening method and is validated in Section 4, Validations. ]]

*Line 118 "Similarly, when the dipole axis and the spin axes are near parallel"*

*I guess "near parallel" should be "near perpendicular" but then I do not see the point.*

**RE: Thank you for the comment. We removed the section on Line 118 causing confusion**

*Line 147 "These contribute to the error on each calculation of the worst-case fields"*

*What is the estimated combined error? The error bars in Figure 5 seem not to include all error contributions. I would expect larger error bars at smaller distances.*

**RE: Thank you for the comment. Estimated combined error comes from intrinsic Twinleaf VMR magnetometer measurement error and horizontal measurement error. Vertical alignment error from a vertical dipole offset cannit be quantified and therefore is not included in estimated combined error. We changed lines 144-147 to reflect this better:**

"The Twinleaf VMR magnetometer has an intrinsic measurement error and there is a horizontal measurement error from centering the object and placing the magnetometer sensors, (which both create the estimated combined error.) There is a vertical alignment error as some objects have non-trivial height which offsets them from the plane of the magnetometer sensors which is set by a finite set of plastic mounts, (which is likely quite small cannot be quantified in this screening process.)"

**While the error bars visually don't appear to grow as they approach the object, we can assure you they do. We quantify estimated combined error as: $\delta B = ( (dB(r)/dr)^2 * (\delta B(r, m)^2 + \delta B\_VMR^2 )^{(1/2)}$. The value for $dB(r)/dr$ has a proportional $1/r^4$ power law relationship, meaning the error grows as distance decreases.**